# Definition and Validation of Fast Design Procedures for Seismic Isolation Systems

Marco Furinghetti [1,2]

1 Civil Engineering and Architecture Department, DICAr, University of Pavia, 27100 Pavia, Italy;
marco.furinghetti@unipv.it
2 EUCENTRE Foundation, 27100 Pavia, Italy

**Abstract:** The research on traditional and innovative seismic isolation techniques has grown significantly in recent years, thanks to both experimental and numerical campaigns. As a consequence, practitioners have also started to apply such techniques in real applications, and nowadays, seismic isolation is widespread in regions characterized by a high level of seismic hazard. The present work aims at providing practitioners with very simple procedures for the first design of the isolation devices of a building, according to the most common typologies of isolators: Rubber Bearings, Lead Rubber Bearings and Curved Surface Sliders. Such Fast Design Procedures are based on simplified approaches, and the mechanical properties of the implemented devices can be obtained by assuming a performance point of the overall structural system, namely effective period and equivalent viscous damping. Furthermore, some important parameters are defined, according to the outcomes of a statistical analysis of the test database of the EUCENTRE Foundation in Italy. Finally, results of a validation study have been provided by analyzing a case-study structure through a Multi Degree of Freedom oscillator and a full 3D Finite Element model.

**Keywords:** seismic isolation; fast design procedure; curved surface slider; rubber bearing; lead rubber bearing





## 1. Introduction

The applications of the seismic isolation technique have become widespread in common practice for both new design and retrofit of existing buildings [1–5]. All the numerical and experimental research works developed in recent decades have led to a better understanding of the behavior of a number of isolators, which are able to provide the proper horizontal stiffness for a period of elongation, together with a certain dissipative capacity, for the reduction of displacement demands. As a consequence, the analytical modeling strategies for the available devices have become more and more realistic, since most of them have been accurately calibrated by means of the outcomes of large experimental campaigns [6–11]. Such models can be used for several purposes, such as the numerical assessment of the response of a base-isolated building and vulnerability studies on a specific building typology, rather than the design of a retrofit application of an existing structural system [12–15]. The commonly adopted standard codes worldwide do provide some general definitions for the seismic isolation system, even though specific strategies on the design procedure are not clearly detectable [16,17].

The present research work aims at providing very simple procedures for a fast initial design of the isolation system of a building, according to the most usual typologies of devices adopted in common practice. Namely, Rubber Bearings, Lead Rubber Bearings and Curved Surface Slider isolators have been considered. Few-step procedures are proposed in order to conceive of both mechanical and geometrical properties of all the devices implemented within the isolation layer, guiding practitioners in choosing the proper values of all the main parameters. To do so, the results of a statistical analysis of the experimental

data contained in the wide database of dynamic tests of EUCENTRE Foundation Laboratory in Italy have been provided, which show the actual distributions of certain mechanical properties of isolation devices. Consequently, the proper range of variation for the related parameter can be considered.

Finally, the proposed Fast Design Procedures were applied to a case-study structure for the sake of validation. Three individual isolation systems were designed, and the efficiency was assessed by performing Non-Linear Time History Analyses. Results showed good agreement between the mean and the single-event demands and the target design values for both the isolation displacement and the building-base shear responses.

## 2. Definition of Fast Design Procedures

In this work, Fast Design approaches have been defined for the definition of the geometrical and mechanical properties of isolation devices. The overall procedures have been developed, considering few steps characterized by very simple assumptions. The initial step in all cases is represented by the choice of the "Performance Point" of the isolation system, which is univocally determined by two important parameters:

- The Design Period $T_d$, which corresponds to the secant period at maximum displacement;
- The Design Equivalent Viscous Damping of the isolation systems $\xi_d$.

According to the aforementioned parameters, the overall base-isolated structural system is considered as an equivalent Single Degree of Freedom (SDOF) oscillator, even though the implemented devices provide a non-linear and hysteretic response. It is well known that standard codes worldwide allow for linear modeling of the isolation system under specific conditions, which are generally related to the energy-dissipation capacity and the force response characteristics of the isolators. Nonetheless, in both numerical and experimental recent works, it has been proved that, for initial design purposes, the behavior of a SDOF oscillator provides a very good estimate of the global behavior of the base-isolated structure. Thus, according to the seismic hazard of the considered construction site, which is represented by the displacement and acceleration response spectra, the performance point is consequently determined in terms of displacement demand and peak normalized force response (normalized with respect to the total mass of the overall system). In what follows, independent procedures have been defined for the most common devices adopted for seismic isolation.

### 2.1. Low and High Damping Rubber Bearings (LDRB & HDRB)

Rubber bearings represent one of the very first typologies of devices capable of protecting the buildings against earthquake excitations, and generally can be associated with linear behavior and with a constant equivalent viscous damping ratio, which can be considered independent with respect to the applied deformation. Figure 1 shows the general composition of Rubber Bearing isolators.

The elastomeric layers provide the low stiffness of the overall device, which will be responsible for the period shift of the base-isolated structure, whereas the thin steel layers lead to high vertical stiffness and stability.

Thus, the first design parameter which can be computed is the height of the device $h_{is}$, assuming the design shear strain of the rubber isolator $\gamma_d$, commonly assumed as 100%. Consequently, given the displacement demand $D_d$, as a function of the displacement spectral coordinate $Sd(T_d, \xi_d)$ at the design period, by accounting for the design equivalent damping ratio:

$$h_{is} = \frac{Sd(T_d, \xi_d)}{\gamma_d} = \frac{D_d}{\gamma_d} \tag{1}$$

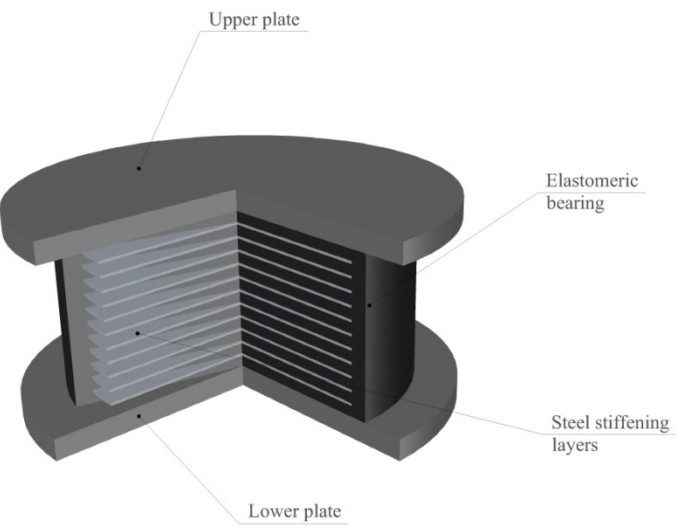

**Figure 1.** Internal components of Rubber Bearing (RB) devices.

The equivalent viscous damping ratio is a constant value, independent with respect to the applied deformation, and is strictly related to the adopted elastomeric compound; namely, LDRB (Low Damping). This is generally characterized by ratios bounded between 5% and 10%, whereas for HDRB (High Damping), higher values from 10% to 15% can be achieved.

In order to determine the horizontal sizes of the device, by assuming a cylindrical isolator, the diameter of the plan section $\Phi_r$ can be computed by scaling the design displacement demand by a proper scale factor, in order not to have buckling behavior at the maximum deformation and to keep the application point of the vertical load displaced inside the base of the device. A multiplication factor equal to 2.0 can be adopted, which leads to correct definitions of both the shape factors $S_1$ and $S_2$ [2]. Hence:

$$\Phi_r \approx 2 \cdot D_d \tag{2}$$

At this step, the device is geometrically defined: the total height of the device can be subdivided into rubber and steel layers (by considering thickness of $10 \div 15$ mm and $2 \div 3$ mm, respectively). Consequently, the lateral stiffness of the single bearing $k_{is}$ can be computed as a function of the geometry (height $h_{is}$ and rubber area section $A_r$) and the shear modulus $G_r$ of the elastomeric compound.

$$k_{is} = \frac{G_r \cdot A_r}{h_{is}} \tag{3}$$

Finally, the number of isolators $n_{is}$ is determined by the ratio between the global stiffness value of the isolation system $K_d$ and the stiffness of the single device $k_{is}$.

$$n_{is} \approx \frac{K_d}{k_{is}} \tag{4}$$

Depending on the initial assumption of the performance point, in some applications, the number of isolation devices could be lower than the total number of bearing points of the base-isolated structure; thus, the isolation system will be consequently constituted of both rubber bearings and low-friction Flat Slider (FS) devices. The low frictional properties of the flat sliding devices allow us not to mismatch the real response of the isolated structural system, in comparison with the assumed performance at the beginning of the design procedure. More specifically, the plane spatial configuration of the devices can be obtained by installing Rubber Bearings along the perimeter of the structure: since flat sliders are supposed to have negligible force response, the designed isolators provide the

maximum contribution in the evaluation of the torsional stiffness of the isolation layer, if the distance between them and the stiffness centroid is maximized.

### 2.2. Lead Rubber Bearings (LRB)

Lead Rubber Bearing devices represent an improved version of the previously analyzed Rubber Bearing isolators. The most important difference is provided by an internal lead core (Figure 2), which leads to much higher dissipative capacity and a non-linear and hysteretic force response.

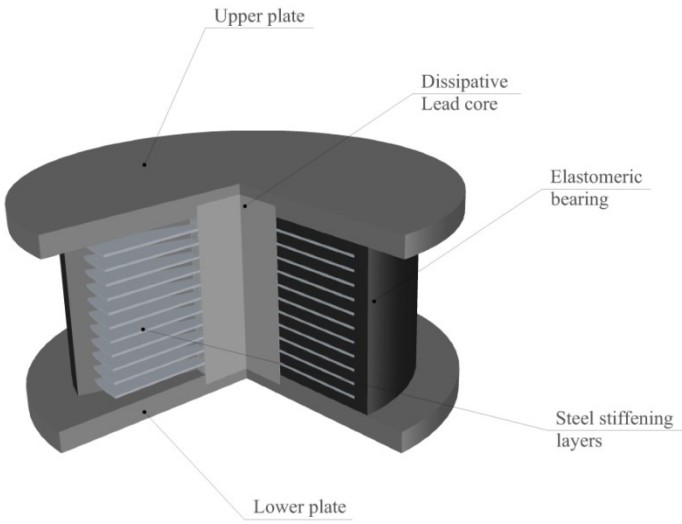

**Figure 2.** Internal components of Lead Rubber Bearing (LRB) devices.

Additionally, for Lead Rubber Bearing devices, the total height of the isolator $h_{is}$ is computed as a function of the desired shear strain $\gamma_d$, commonly assumed around 100%.

$$h_{is} = \frac{Sd(T_d, \xi_d)}{\gamma_d} = \frac{D_d}{\gamma_d} \qquad (5)$$

Since the superposition of the effects of the lead core (elastoplastic hysteretic rule) and the rubber portion of the device (linearly modeled) leads to an elastoplastic with linear hardening behavior, the main mechanical properties are represented by the yielding displacement and strength and the post-yield stiffness. In order to provide practitioners with the ordinary range of the shear strain of the lead core, which can be experienced in common practice, the huge dataset of dynamic tests performed at the Laboratory of EUCENTRE Foundation was analyzed [18]. More specifically, for thousands of tests, all the hysteretic loops were bi-linearized, and characteristics of the obtained bi-linear approximation were statistically studied. In Figure 3, results in terms of experimental variability of the yielding shear strain $\gamma_y$ for the lead core are reported.

From a qualitative perspective, the overall variability could be associated with a log-normal probability density function, with a mode value corresponding to approximately 2.5% of yielding shear strain of the lead core, assumed as the ratio between the yielding displacement and the total height of the device. Thus, for design purposes, such a value can be assumed as a reference; consequently, the yielding displacement of the isolator $D_y$ is computed.

$$\begin{cases} \gamma_y \approx 2.5\% \\ D_y = \gamma_y \cdot h_{is} \end{cases} \qquad (6)$$

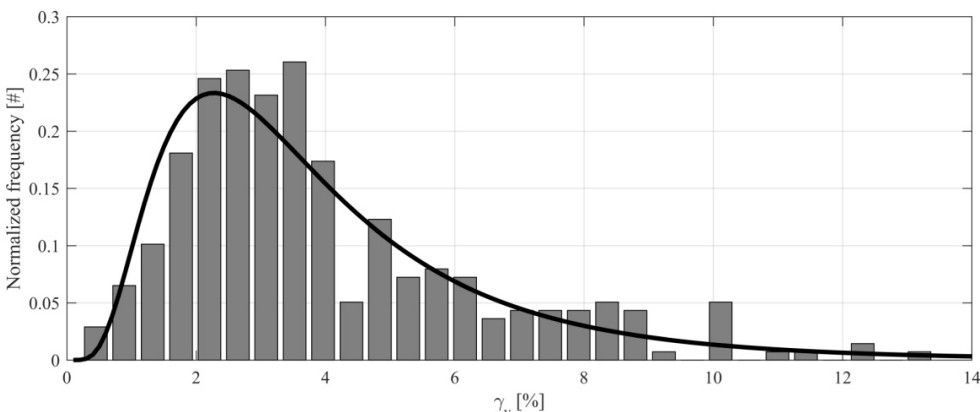

**Figure 3.** Statistical variability of the experimental shear strain for LRB devices.

In order to proceed with the next design steps, the idealized bi-linear behavior of the device is analyzed in a normalized way: namely, the horizontal displacement axis of the force-displacement hysteretic response is normalized with respect to the yielding displacement, whereas the vertical force axis values are divided by the yielding strength of the lead core (Figure 4). Consequently, the point at the peak deformation of the isolator is represented by two important parameters: the ductility demand $\widetilde{\mu}$ and the overstrength ratio $\alpha$.

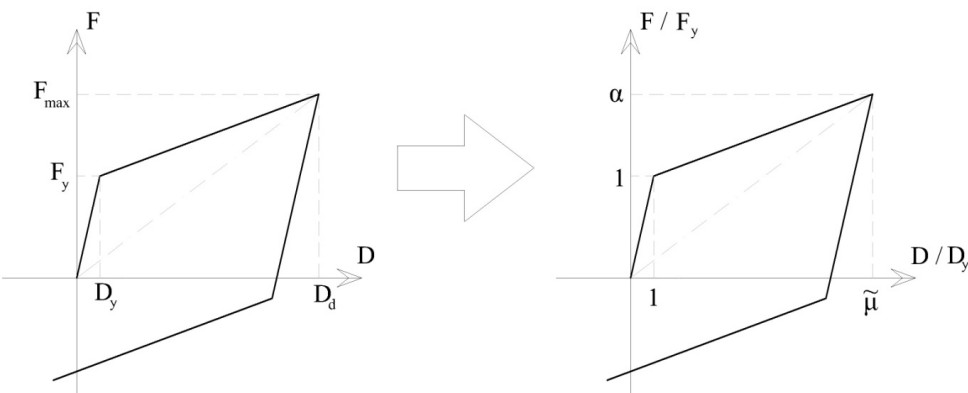

**Figure 4.** Normalized behavior for LRB devices.

The relationship between these fundamental mechanical properties can be found by imposing the initial damping value, which is assumed for the definition of the performance point of the isolation system. For LRB devices, the damping ratio is no longer constant with respect to deformation, given the highly non-linear hysteretic consecutive law, and consequently, the Jacobsen formulation of the equivalent viscous damping has to be considered. Thus, for the computation of the damping ratio, the Energy Dissipated per Cycle (*EDC*) is computed, according to a symmetric hysteresis cycle in the design displacement.

$$\begin{cases} EDC = 4 \cdot (\widetilde{\mu} - \alpha) \cdot D_y \cdot F_y \\ \\ \zeta_d = \dfrac{EDC}{2\pi \cdot D_d \cdot F_{max}} = \dfrac{2 \cdot (\widetilde{\mu} - \alpha)}{\pi \cdot \widetilde{\mu} \cdot \alpha} \end{cases} \tag{7}$$

Being $F_{max}$ the maximum force of the device in the design displacement. The aforementioned definition allows us to express the overstrength ratio of the device as a direct function of the ductility demand and the design equivalent viscous damping ratio, as shown in the following expression:

$$4 \cdot (\widetilde{\mu} - \alpha) \cdot D_y \cdot F = 2\pi \cdot D_d \cdot F_{max} \cdot \zeta_d \tag{8}$$

Hence, since the ductility demand can be computed as the ratio between the assumed values for design and yielding shear strains, the overstrength ratio is univocally determined.

$$\begin{cases} \widetilde{\mu} = \dfrac{D_d}{D_y} = \dfrac{\gamma_d}{\gamma_y} \\[2ex] \alpha = \dfrac{2\widetilde{\mu}}{2 + \pi \cdot \widetilde{\mu} \cdot \zeta_d} \end{cases} \tag{9}$$

For the definition of the geometry of the internal lead core, two additional parameters have been statistically analyzed within the outcomes of the dynamic tests of the EUCENTRE Foundation database: namely, the yielding stress $\tau_y$ and a shape parameter of the lead core. The latter parameter is defined as the ratio between the diameter of the lead core $\Phi_l$ and the total height of the device. In Figure 5, statistical results are shown.

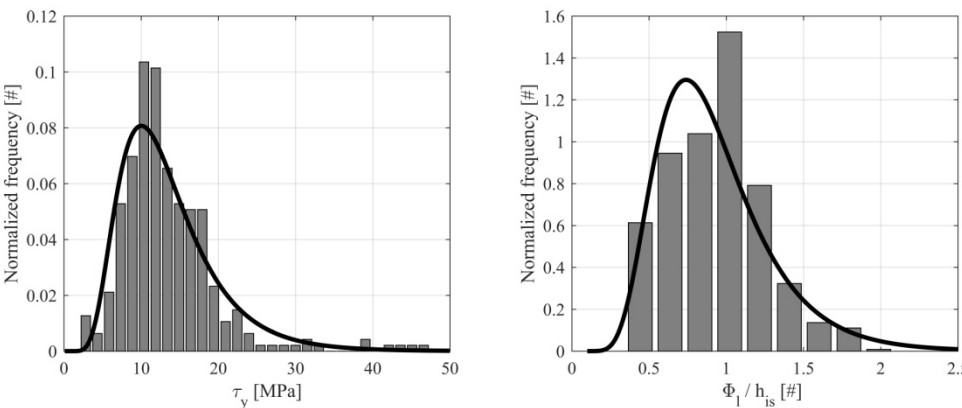

**Figure 5.** Statistical distributions for the yielding stress and the shape parameter of the lead core.

Graphical results for these parameters could still be addressed to a log-normal distribution, as previously noted for the yielding shear strain of the lead core. In order to validate such statistical results, a non-linear best-fit procedure has been applied by considering the analytical equation of a log-normal distribution for all variables. Graphical results are shown in Figure 6, together with the $R^2$ value returned by the non-linear best-fit procedure.

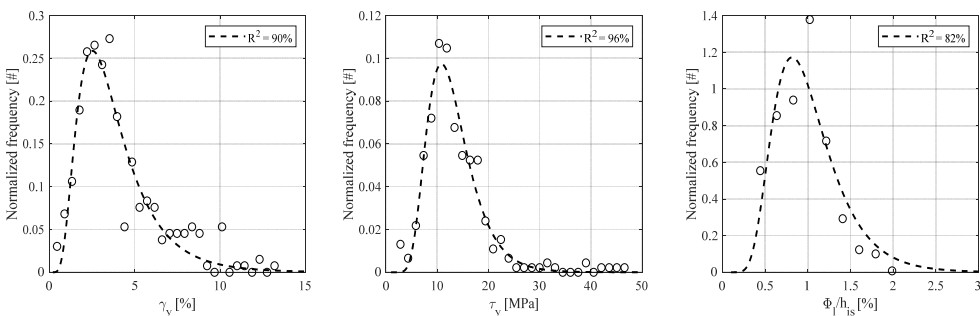

**Figure 6.** Validation of statistical results.

As can be noted, for all parameters, the $R^2$ value is greater than 80%; consequently, the adoption of log-normal distributions seems reasonable. In Table 1, the main characteristics of the obtained distributions are listed in terms of mean and standard deviation of the correspondent Gaussian distribution of the natural logarithm of all variables, together with Mean, Median and Mode values of the overall probability density functions.

**Table 1.** Characteristics of the obtained distributions.

|  | μ | σ | Mean | Median | Mode |
|---|---|---|---|---|---|
| $\gamma_y$ [%] | 1.208 | 0.532 | 3.9 | 3.3 | 2.5 |
| $\tau_y$ [MPa] | 2.509 | 0.355 | 13.1 | 12.3 | 10.8 |
| $\Phi_l/h_{is}$ [#] | −0.054 | 0.388 | 1.0 | 0.9 | 0.8 |

It has to be noted that the proposed numerical values in the presented design procedure correspond to the most common values adopted in real applications, generally referred to as the mode value, as the most recurrent configuration. On the other hand, results also show that the variability of such parameters is not negligible, and consequently, random variables could be actually considered if bound analyses are needed for a more comprehensive evaluation of all the worst scenarios. Consequently, the yielding force of the device can be computed as follows:

$$\begin{cases} \Phi_l \approx 0.8 \cdot h_{is} \\ \tau_y \approx 11 MPa \\ F_y = \tau_y \cdot A_l \end{cases} \tag{10}$$

Hence, all the remaining parameters can be easily computed. The peak force $F_{\max}$ of the device is then defined as the yielding force of the lead core, scaled by the overstrength ratio.

$$F_{\max} = \alpha \cdot F_y \tag{11}$$

Thus, the secant stiffness of the single device at maximum displacement $k_{is}$ can be determined as the ratio between the peak force response and the design displacement:

$$k_{is} = \frac{F_{\max}}{D_d} \tag{12}$$

Finally, the number of needed isolation devices $n_{is}$ is again represented by the ratio between the global stiffness $K_{tot}$ of the isolation system and the secant stiffness of the single isolator $k_{is}$.

$$n_{is} \approx \frac{K_{tot}}{k_{is}} \tag{13}$$

The last parameter is represented by the external diameter of the horizontal section of the isolator $\Phi_r$, which can be computed by considering the post-yield stiffness as the contribution of the linearly modeled rubber portion of the device. Thus, the area of the rubber portion of the device $A_r$ can be computed and, consequently, the external diameter of the device is obtained by considering the previously defined diameter of the lead core.

$$\frac{G_r \cdot A_r}{h_{is}} = \frac{F_{\max} - F_y}{D_{\max} - D_y} \Rightarrow A_r = \frac{h_{is}}{G_r} \cdot \frac{F_{\max} - F_y}{D_{\max} - D_y} \Rightarrow \Phi_r \tag{14}$$

As observed for purely rubber bearings, if the number of needed devices is lower than the total number of bearing points of the system, Flat Sliders with low frictional response have to be adopted.

*2.3. Curved Surface Sliders (CSS)*

Nowadays, Curved Surface Slider devices are widely used in worldwide applications of common practice. Such isolators, if properly designed, can accommodate large displacement demands, and high dissipative capacities can be achieved, thanks to the frictional motions along the spherical surfaces. In addition, a certain recentering behavior is ensured due to the stepwise projection of the applied vertical force on the horizontal plane during motion. In Figure 7 an example of a CSS isolator is shown, equipped with two spherical sliding surfaces and an internal slider, characterized by a unique steel block.

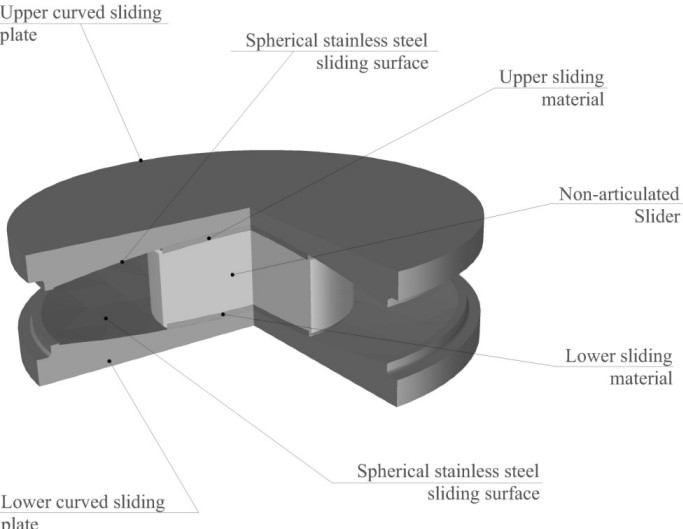

**Figure 7.** Internal components of a Double Curved Surface Slider isolator.

The main mechanical properties of the devices are represented by the equivalent radius of curvature and the design value of the friction coefficient. Such parameters determine the hysteretic non-linear behavior of the devices, which implies a deformation-dependent equivalent damping ratio. Thus, as for the LRB case, the Jacobsen formulation of the equivalent viscous damping ratio is needed [19], according to a symmetric full cycle at the design displacement. In addition, an average friction coefficient per cycle can be defined, as ruled by the European standard code for Anti-Seismic devices UNI:EN15129:2009: both of the aforementioned quantities are direct functions of the Energy Dissipated per Cycle (*EDC*).

$$\begin{cases} \xi_d = \frac{EDC}{2\pi \cdot D_d \cdot F_{\max}} \\[2mm] \mu_d = \frac{EDC}{4 \cdot W \cdot D_d} \end{cases} \tag{15}$$

By collecting the *EDC* parameter from the first equation, the expression of the design friction coefficient $\mu_d$ can be found as a function of the design equivalent viscous damping ratio $\xi_d$, the acceleration spectral coordinate *Sa* $(T_d, \xi_d)$ of the chosen performance point and the maximum equivalent damping ratio related to hysteretic behaviors $\xi_{lim}$, which is equal to $2/\pi$ (about 64% [19]).

$$\mu_d = \frac{\xi_d}{\xi_{\lim}} \cdot \frac{Sa(T_d, \xi_d)}{g} \tag{16}$$

Finally, the definition of the maximum force at the design displacement, normalized with respect to the total weight of the overall base-isolated structure *W*, leads to the analytical expression of the equivalent radius of curvature of the device $R_{eq}$.

$$\begin{aligned} \frac{F_{\max}}{W} &= \frac{Sa(T_d, \xi_d)}{g} = \mu_d + \frac{D_d}{R_{eq}} \\ \frac{Sa(T_d, \xi_d)}{g} &= \frac{\xi_d}{\xi_{\lim}} \cdot \frac{Sa(T_d, \xi_d)}{g} + \frac{Sd(T_d, \xi_d)}{R_{eq}} \end{aligned} \tag{17}$$

Hence:

$$R_{eq} = \frac{g}{\omega_{is}^2} \cdot \frac{\xi_{\lim}}{\xi_{\lim} - \xi_d} \tag{18}$$

Concerning the geometrical characteristics of the device, the typology of the CSS isolator has to be assumed: namely, a Single Curved rather than Double Curved Surface Slider (SCSS or DCSS). According to such assumption, the overall geometrical parameter can be detected. In addition, the horizontal diameter of the internal sliding interfaces is

strictly related to the response of the implemented sliding material, and more specifically to the dependency of the friction coefficient with respect to the contact pressure.

## 3. Case-Study Structure for Validation Analyses

In order to validate the effectiveness of the proposed Fast Design Procedures, a case-study structure was considered as an existing building needing a retrofit application for seismic vulnerability reduction (Figure 8).

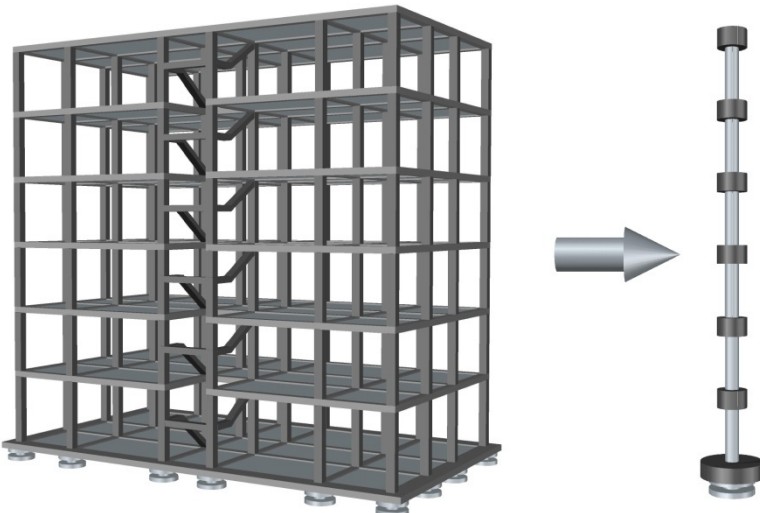

**Figure 8.** Case study structure used for validation results.

The structural system consists of a six-story reinforced concrete frame building, with all members designed according to Italian code-conforming provisions [20]. The plan dimensions are approximately 21 m and 12 m for the x and y directions, respectively, and the inter-story height is 3.05 m for all the floors but the ground one, which is 3.4 m height, with a consequent total height equal to 19 m.

At each story of the system, the seismic mass is approximately equal to 300 tons, whereas the total weight of the structure is 2080 tons. If linear elastic behavior is considered for all structural members, the first mode of vibration of the structure is represented by a period around 1.0 s. On the other hand, in order to provide a comparison between the base shear demand and capacity of the building equipped with the designed isolation systems, a non-linear static analysis (push-over) was computed, according to an OpenSees model of the fixed-base configuration. In Figure 9, results are shown in terms of the capacity curve of the structure.

The ultimate conditions of the building were detected by considering a reduction in the maximum strength of the curve of 20%. Consequently, a bi-linear approximation of the non-linear capacity curve was computed through a special least-square procedure in order to minimize the sum of square errors of the base shear values, ensuring the same area below each curve as an energy balance. Hence, the maximum strength of the building at the yielding point of the bi-linear curve was equal to 2862 kN.

The response of the designed base-isolated buildings was computed by using two individual models:

- A Full-3D model, implemented in the F.E.M. commercial software SAP2000 [21], accounting for linear-elastic beam and columns, with proper non-linear links for isolation devices;
- A statically condensed Multi Degree of Freedom (MDOF) oscillator, with the same dynamic properties of the aforementioned F.E.M. model [22,23].

The simplified dynamic system of the building, represented by the lumped mass oscillator, was not to be intended as a conservative strategy for the evaluation of the base-

isolated structure, but just as a faster alternative in comparison to the more detailed full F.E.M. model.

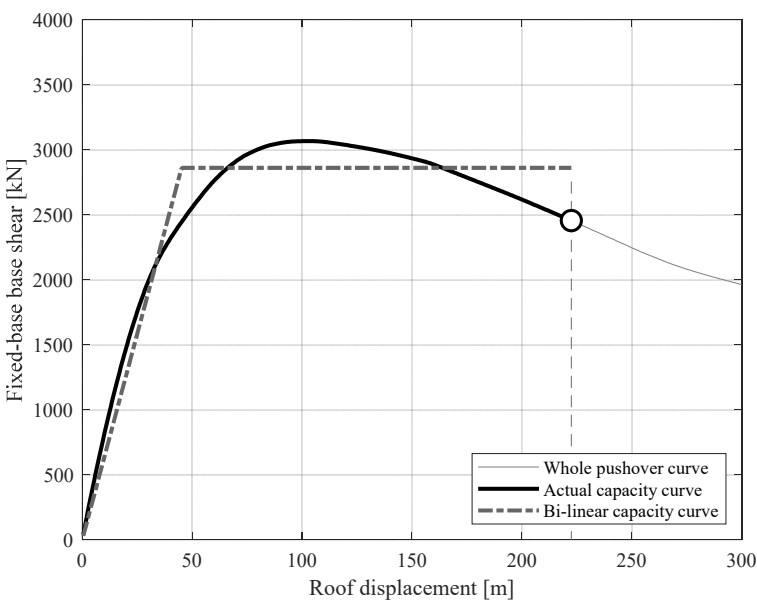

**Figure 9.** Non-linear capacity curve of the case study structure—fixed-base configuration.

### 3.1. Definition of the Seismic Input

In this research work, validation results were obtained through Non-Linear Time History Analyses (NLTHA), by considering a selection of seven unidirectional natural seismic events [24]. According to the Italian Building Code, the spectrum compatibility of the selected set of ground motions was studied by considering the seismic hazard level defined for the construction site: more specifically, L'Aquila was considered as a reference location, with Soil class C and topography category T1. The Collapse Limit State was considered, which is related to 5% probability of exceedance within the reference life of the building, assumed as 50 years (which implies a return period of 975 years). Thus, the spectrum compatibility was checked by considering lower and upper bounds for the mean spectrum of the selected events as 90% and 130%, respectively, of the code-target spectrum, and a period range within 0.15 s and 3.0 s was considered. Individual ground motion records were scaled in order to better achieve spectrum-compatibility prescriptions; moreover, scale factors were bounded between 0.5 and 2.0, aiming at preserving the correct frequency content for the considered Peak Ground Acceleration (PGA) values. Table 2 summarizes the main characteristics of the selected records, whereas Figure 10 provides graphical representation of the spectrum-compatibility check.

Results of the spectrum-compatibility check are shown, in terms of individual and mean response spectra, in comparison to the target, lower- and upper-bound graphs.

Thanks to a special selection procedure, the adopted set of ground motion was characterized by both a mean and a single-event good matching with respect to the target spectrum, and consequently limited inter-event variability was expected in the final results.

The selected earthquake excitations were applied along a single direction of the models, corresponding to the *x*-axis, which is parallel to the longest size of the building.

**Table 2.** Characteristics of the selected seismic events.

| Event [#] | Station ID | Earthquake Name | Date | Mw | Fault Mechanism | Epicentral Distance [km] | Original PGA [g] | Scaled PGA [g] | Scale Factor [#] |
|---|---|---|---|---|---|---|---|---|---|
| 1 | ST164(x) | Kalamata | 13/09/1986 | 5.9 | normal | 10.0 | 0.215 | 0.429 | 2.00 |
| 2 | ST163(x) | Kalamata | 13/09/1986 | 5.9 | normal | 11.0 | 0.240 | 0.479 | 2.00 |
| 3 | ST271(y) | Dinar | 01/10/1995 | 6.4 | normal | 8.0 | 0.319 | 0.404 | 1.27 |
| 4 | ST561(x) | Izmit | 17/08/1999 | 7.6 | strike slip | 47.0 | 0.238 | 0.475 | 2.00 |
| 5 | EC04(y) | Imperial Valley | 15/10/1979 | 6.5 | strike-slip | 27.0 | 0.485 | 0.485 | 1.00 |
| 6 | EC05(y) | Imperial Valley | 15/10/1979 | 6.5 | strike-slip | 27.7 | 0.519 | 0.519 | 1.00 |
| 7 | ERZ(x) | Erzincan | 13/03/1992 | 6.6 | strike-slip | 9.0 | 0.495 | 0.446 | 0.90 |

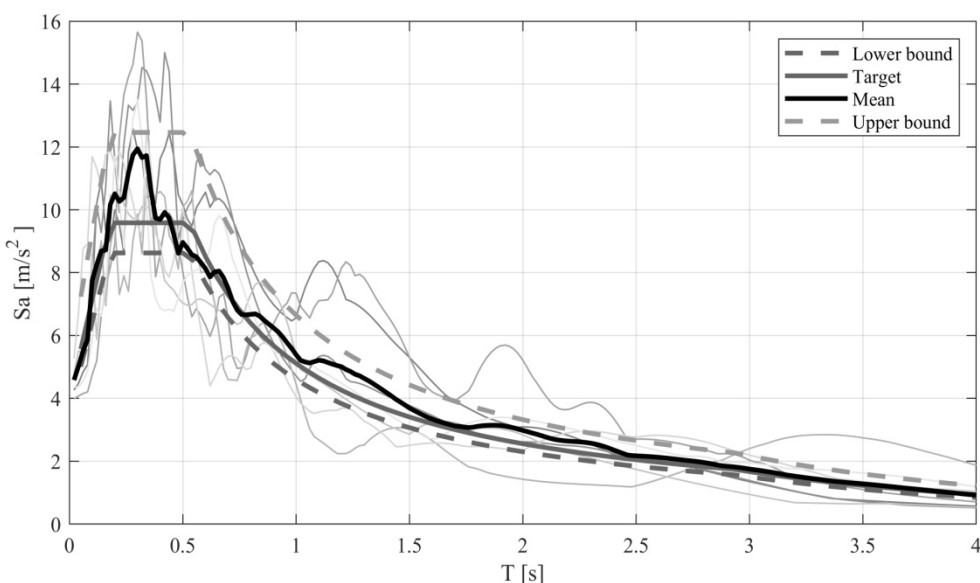

**Figure 10.** Code-conforming spectrum-compatibility check of the selected records.

*3.2. Designed Isolation Systems*

Three independent isolation systems were designed according to the Fast Design Procedures presented in this work. More specifically, High Damping Rubber Bearings and Lead Rubber Bearings were designed, together with Flat Sliders, in order to provide the proper number of total bearing points, whereas Curved Surface Slider devices could be installed in all locations. In Tables 3–5 results of the design procedures are provided for RB, LRB and CSS isolators, respectively.

**Table 3.** Design procedure results—HDRB.

| $T_d$ [sec] | $\xi_d$ [%] | $h_{is}$ [mm] | $\gamma_d$ [%] | $G_r$ [MPa] | $\Phi_r$ [mm] | $n_{is}$ [mm] |
|---|---|---|---|---|---|---|
| 2.8 | 15 | 256 | 100% | 1 | 512 | 12 |

**Table 4.** Design procedure results—LRB.

| $T_d$ [sec] | $\xi_d$ [%] | $h_{is}$ [mm] | $\gamma_d$ [%] | $G_r$ [MPa] | $\Phi_l$ [mm] | $\Phi_r$ [mm] | $n_{is}$ [mm] |
|---|---|---|---|---|---|---|---|
| 2.5 | 30 | 169 | 105% | 1 | 136 | 469 | 8 |

**Table 5.** Design procedure results—CSS.

| $T_d$ [sec] | $\xi_d$ [%] | $\mu_d$ [%] | $R_{eq}$ [m] |
|---|---|---|---|
| 2.7 | 30 | 4.97 | 3.413 |

For elastomeric bearings, isolation devices were placed at the edge of the perimeter of the building in order to maximize the torsional strength of the isolation system, and Flat Sliders were implemented by considering a friction coefficient of 1.0%.

### 3.3. Validation Results

In Figure 11, an example of hysteresis loops of the whole isolation system was shown for both the F.E.M. and the MDOF models.

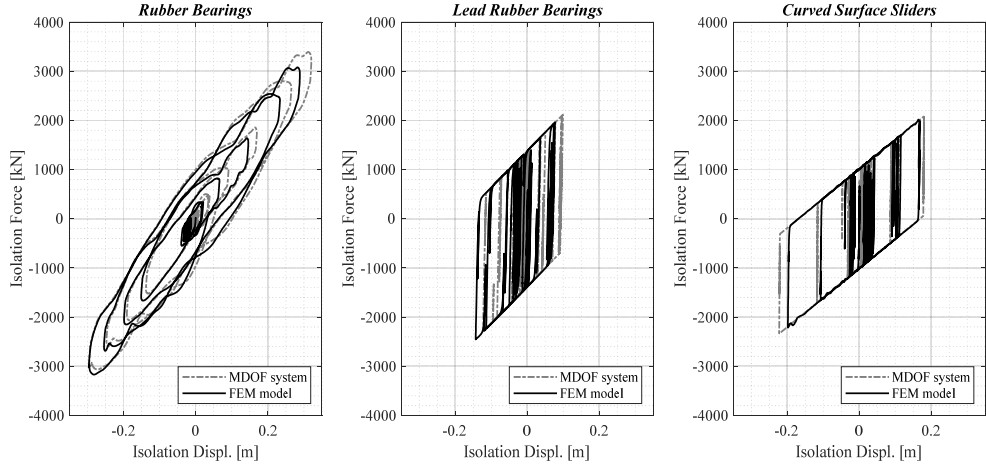

**Figure 11.** Example of hysteresis loops of the designed isolation systems.

As can be seen, both the models provide almost overlapped hysteretic behaviors, which suggests a good comparison between simplified rather than detailed modeling strategies results. Moreover, it is possible to note that the proper non-linear behavior of all the isolation technologies adopted in this work is fairly represented by the implemented models.

Finally, the results of the efficiency of the designed isolation systems are analyzed in Figures 12 and 13, in terms of single-event and mean response for peak isolation displacement and building base-shear, respectively.

As can be noted, for all the typologies of isolation bearings analyzed in this work, the mean peak displacement demand results were lower than or approximately equal to the design displacement, returned by the initial assumption of the performance point. In addition, even if the single-event response is analyzed, fairly good results were noticed, since the maximum variation with respect to the design value is about +25%, which corresponds to a generally allowable extra-displacement capacity for all the typologies of isolators.

Concerning the building base-shear response, all values resulted as lower than or approximately equal to the strength of the fixed-base bi-linear capacity curve: thus, the building is properly protected by the designed isolation systems, and the seismic vulnerability is consequently reduced. This can be noticed not only for the mean response, but also for the majority of the single-event response.

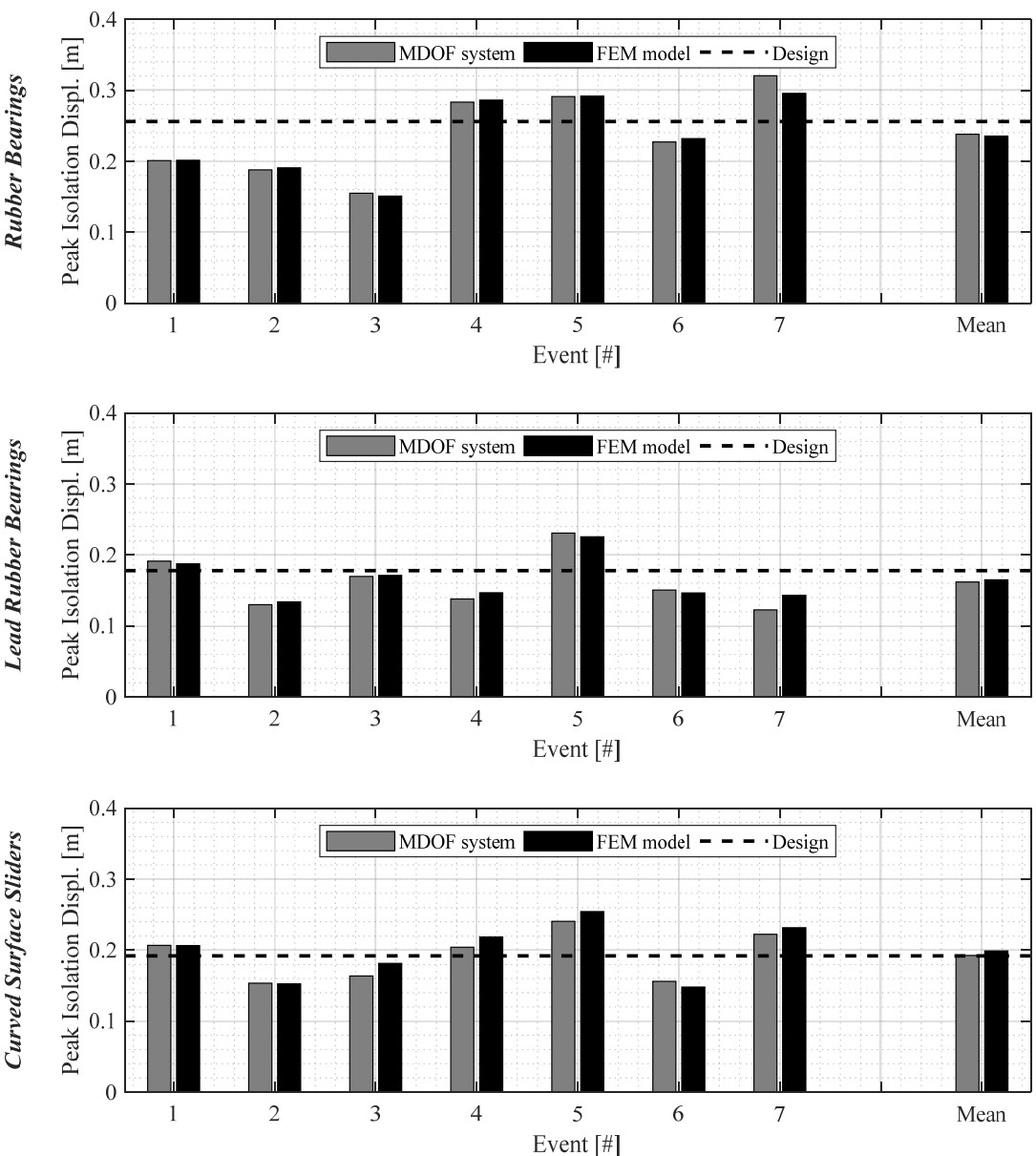

**Figure 12.** Validation results—peak displacement response for RB (**top**), LRB (**middle**) and CSS (**bottom**) devices.

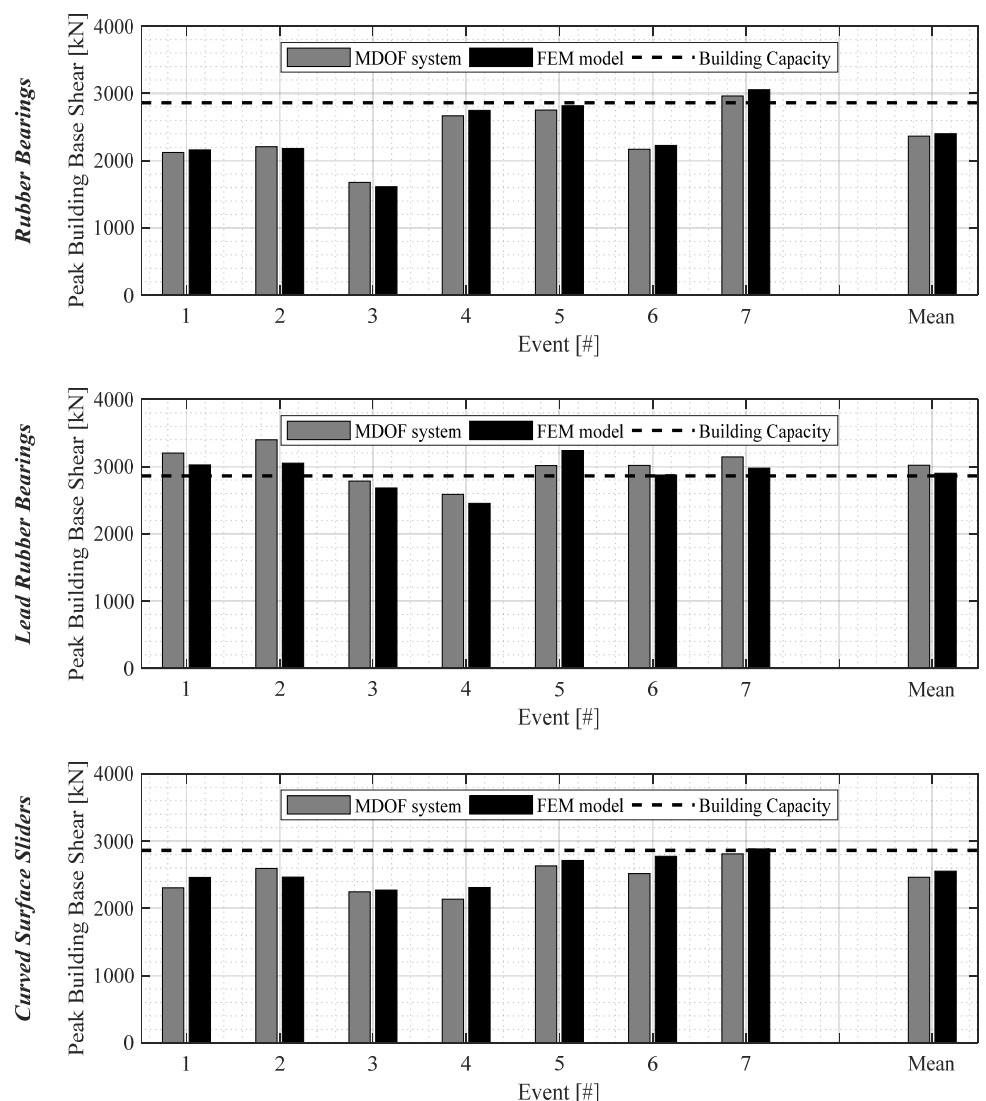

**Figure 13.** Validation results—peak building base-shear response for RB (**top**); LRB (**middle**); and CSS (**bottom**) devices.

## 4. Discussion

The proposed procedures are able to provide an initial set of both mechanical and geometrical design parameters, and the validation results shown in this work suggest that the implemented bearings are effectively able to protect the superstructure, leading to the desired performances. On the other hand, the consistency of the designed characteristics have to be checked in detail, and also according to the results returned by Non-Linear Time History Analyses, considering the whole 3D structural system together with the proper non-linear hysteretic behavior. Special attention should be focused on the variation during the motion of the vertical load applied to the devices, together with the achieved peak velocity values, which would be the reference parameter for a consequent testing protocol according to standard codes for anti-seismic devices, such as UNI:EN15129:2009. In other words, the proposed design procedures return an initial configuration of the isolation system, which has to be checked with the proper analysis, by modeling the force response and the hysteretic behavior of the adopted isolators in the most realistic way possible. Finally, all the results obtained in the analyses provide all the key parameters which will be references for the definition of the testing protocol ruled by national standards, with the aim of an experimental assessment of the designed isolation devices.

## 5. Conclusions

This research work provides Fast Design Procedures for isolation devices, which can lead to an efficient definition of both geometrical and mechanical properties of the considered isolation technology. Specifically, the behavior of the most common isolators used in real practice applications was analyzed, and a strategy for the computation of the main response parameters was developed. In addition, statistical analyses of the dynamic tests database of the EUCENTRE Foundation in Italy were reported in order to provide practitioners with some important guidelines for the proper assumption of initial key parameters. All procedures were defined in the easiest way possible, starting from the detection of the desired performance point of the overall base-isolated system, in terms of period and equivalent viscous damping. Finally, the presented procedures were validated by designing three individual isolation systems for a case-study structure, and by computing the response of the overall structures by means of Non-Linear Time History Analyses. Results have shown the effectiveness of the designed isolation systems from both the isolator peak displacement demand and the building base-shear response.

**Funding:** This study was funded by Italian Civil Protection (Convenzione DPC-EUCENTRE 2017-2019-ReLUIS project 2019–2021).

**Data Availability Statement:** Data sharing is not applicable to this article.

**Acknowledgments:** Part of the current work was carried out under the financial support of the Italian Civil Protection, within the frameworks of the Executive Project 2017–2019 (Project 3—Assessment of the seismic isolation of building structures through hybrid tests with numerical substructuring) and the national Research Project DPC-ReLUIS (National Network of Laboratories of Seismic Engineering) 2019–2021, WP15.

**Conflicts of Interest:** The author declares no conflict of interest.

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
