# Peer review of "Definition and Validation of Fast Design Procedures for Seismic Isolation Systems"

_vibration, doi:10.3390/vibration5020017_

Round 1
Reviewer 1 Report
Please consider the comment in attached file

Author Response
The Author wish to thank the Distinguished Reviewer for the precious comment on this important research work. All the revisions are commented in the attached pdf file.

Reviewer 2 Report
The article is an overview of mechanical devices designed to isolate buildings from the effects of seismic waves. These devices use different principles to isolate exposure, which can be conditionally divided into 2 types: vibration damping using rubber bearings and the use of movements on curved surfaces with increased friction. Quantitative estimates of the degree of damping of the oscillation energy in these devices are presented. Frequency-dependent estimates of the degree of vibration damping are given. The question of the optimal choice of the parameters of various damping devices and their number to ensure the maximum damping of oscillations for the most critical frequencies of buildings is discussed. An example of evaluating the effectiveness of various methods of isolating seismic effects is considered on the example of a model of a building structure, for which it is necessary to carry out work on its seismic strengthening. Seismic effects were simulated using a special mechanical oscillator. Seismic effects were selected so that their parameters were similar to seismic vibrations for 7 earthquakes in different regions of the world. For the considered building model, estimates of the efficiency of vibration damping for various methods of seismic isolation were obtained.
Author Response
The Author wish to thank the Distinguished Reviewer for the comments on this important manuscript. The main objectives of the manuscript have been clearly understood by the Reviewer in all the provided details, which suggest that the descriptions of the presented procedures and the discussion on the obtained validation results provide the proper information to whom could be in charge of designing an isolation system for buildings.
